# Increased Total Serum Bilirubin Level Post-Ibuprofen Use Is Inversely Correlated with Neonates’ Body Weight

**DOI:** 10.3390/children9081184

**Published:** 2022-08-07

**Authors:** Zon-Min Lee, Yao-Hsu Yang, Ling-Sai Chang, Chih-Cheng Chen, Hong-Ren Yu, Kuang-Che Kuo

**Affiliations:** 1Department of Pharmacy, Kaohsiung Chang Gung Memorial Hospital, Kaohsiung 83301, Taiwan; 2Department of Pharmacy, Tajen University, Pingtung 90741, Taiwan; 3Department of Traditional Chinese Medicine, Chiayi Chang Gung Memorial Hospital, Chiayi 61344, Taiwan; 4Health Informatics and Epidemiology Laboratory, Chiayi Chang Gung Memorial Hospital, Chiayi 61344, Taiwan; 5School of Traditional Chinese Medicine, College of Medicine, Chang Gung University, Taoyuan 33303, Taiwan; 6Department of Pediatrics and Kawasaki Disease Center, Kaohsiung Chang Gung Memorial Hospital, Kaohsiung 83301, Taiwan; 7Department of Pediatrics, Chiayi Chang Gung Memorial Hospital, Chiayi 61344, Taiwan; 8College of Medicine, Chang Gung University, Kaohsiung 83301, Taiwan; 9Section of Neonatology, Kaohsiung Chang Gung Memorial Hospital, Kaohsiung 83301, Taiwan; 10Department of Early Childhood Care and Education, Cheng-Shiu University, Kaohsiung 83301, Taiwan; 11Department of Pediatrics, Kaohsiung Chang Gung Memorial Hospital, Kaohsiung 83301, Taiwan

**Keywords:** ibuprofen, hyperbilirubinemia, neonate

## Abstract

Backgrounds: Drugs with the ability to displace bilirubin from albumin-binding sites subsequently leading to an increased bilirubin level may cause hyperbilirubinemia in neonates. Ibuprofen is commonly used to treat patent ductus arteriosus (PDA) in neonates, yet the use of ibuprofen has drawn mixed conclusions. We performed a retrospective study to determine how ibuprofen use influences the total serum bilirubin (TSB) level in neonates of differing birth weight (BW). Materials and methods: Neonates (including premature infants) born at Chang Gung Memorial Hospital, Taiwan during January 2004 to July 2020 were entered into this study. We recorded the phototherapy duration, including the initial day and end day, and determined the average influence of one-day phototherapy on TSB level. The highest monitored TSB level post-ibuprofen use minus the one measured prior to ibuprofen use was considered the TSB change following ibuprofen administration in this study, and the above-mentioned influence of daily phototherapy on the TSB level was used to correlate the results. Neonates with any of the following conditions were excluded: those who received ceftriaxone, those with intraventricular hemorrhage, and those infected with TORCH. Results: The average daily influence of phototherapy on the TSB level of neonates was −0.20 (−0.57~0.05) mg/dL, −0.28 (−0.84~0.13) mg/dL, −0.75 (−1.77~0.10) mg/dL, and −1.60 (−2.70~−0.50) mg/dL in neonates with BWs of <1 kg, 1–1.49 kg, 1.5–2.49 kg, and ≥2.5 kg, respectively, indicating that neonates with a BW ≥ 1.5 kg experienced a greater reduction in TSB level following phototherapy as compared with those with a BW < 1.5 kg. The average TSB increase following ibuprofen use in neonates was 3.38 ± 2.77 mg/dL, 2.04 ± 2.53 mg/dL, and 1.34 ± 2.24 mg/dL in neonates with BWs of <1 kg, 1–1.49 kg, and ≥1.5 kg, respectively, i.e., an elevated TSB change with a decreased neonate BW was noted post-ibuprofen use (*p* = 0.026, one-way analysis of variance (ANOVA)). Conclusions: As ibuprofen use is correlated with an apparent increase in TSB level in neonates with a lower BW, especially in those with a BW < 1 kg, iv acetaminophen can be an appropriate alternative to ibuprofen for ELBW neonates for the treatment of PDA if they are experiencing severe unconjugated hyperbilirubinemia.

## 1. Introduction

Neonatal hyperbilirubinemia, an increased total serum bilirubin (TSB) level, is a common occurrence in neonates [1]. Bilirubin, a product of hemoglobin degradation [2], is transported in an unconjugated form, largely bound to albumin in the blood, and the liver transforms it into conjugated bilirubin, which is then excreted in bile [1]. Hyperbilirubinemia is caused by increased bilirubin deposition in the tissue, causing yellow pigmentation of the skin [1], as bilirubin production surges by thrice in neonates in comparison with adults, mostly owing to a neonate’s reduced red blood cell (RBC) lifespan and an impaired ability to eliminate bilirubin [3].

Blood incompatibility and glucose-6-phosphate dehydrogenase (G6PD) deficiency have been implicated in the development of neonatal hyperbilirubinemia [4], and hemolytic disease of the newborn (HDN) may lead to severe hyperbilirubinemia. [5] The etiology of HDN starts with the attack of fetal RBCs by maternal antibodies owing to incompatibility of fetal and maternal blood attributed to the Rhesus and ABO antigen systems [5]. Hyperbilirubinemia that continues without appropriate management may result in harmful effects including jaundice, neurotoxicity, and brain dysfunction, and may also cause pharmacokinetic alterations of drugs [6,7].

Unbound (free) bilirubin has a better sensitivity and specificity than TSB and can predict the risk of bilirubin neurotoxicity more precisely [8]. Free bilirubin is capable of penetrating the blood–brain barrier [9], and in serum is controlled by albumin binding due to both a large quantity of albumin and its affinity for bilirubin [7]. The binding of bilirubin to albumin is reversible and quick, involving a dynamic equilibrium with bilirubin continuously binding with albumin and then separating [7]. However, a method of measuring free bilirubin is not widely available, thus suggesting that clinicians ought to consider methods other than this with regard to therapeutic decisions. 

Although not useful as a sensitive and specific predictor of neurological outcomes, and poorly correlated with bilirubin neurotoxicity [9], TSB is nowadays regularly used to guide treatment post-phototherapy for hyperbilirubinemia in neonates. TSB indicates the risk of neurological damage regarding the beginning or ending of phototherapy treatment, or when to initiate exchange transfusions if necessary [10]. Clinical drugs with the ability to displace bilirubin from albumin-binding sites, subsequently leading to increased TSB, have been associated with certain causes of hyperbilirubinemia in neonates, especially in those with certain diseases or genetic variations [6,11]. 

Ibuprofen is known to inhibit prostaglandin synthesis, although the mechanism of action by which ibuprofen closes the patent ductus arteriosus (PDA) is still undetermined, and this drug is commonly used for the treatment of PDA in neonates [12]. However, the use of ibuprofen has drawn mixed conclusions, with one study [13] reporting that it is associated with a higher peak TSB level, and another study [14] indicating that ibuprofen may not be associated with the bilirubin displacement effect in relatively stable premature infants with mild to moderate unconjugated hyperbilirubinemia.

As TSB, the most constantly monitored index for hyperbilirubinemia, is poorly correlated with bilirubin neurotoxicity, we performed a retrospective study to determine how ibuprofen use influences the TSB level in neonates of differing birth weight (BW) and gestational age (GA).

## 2. Materials and Methods

### 2.1. Data Source

The present study utilized the birth record files of Chang Gung Research Database (CGRD), encompassing all neonates (including premature infants) born at Chang Gung Memorial Hospital (including multiple centers), Taiwan (including those being hospitalized, or in an outpatient clinic or the emergency department). The study period was between January 2004 and July 2020. All neonates were recorded within 30 days after birth. Those who were administered ibuprofen iv or via oral solution (10 mg/kg, followed by 5 mg/kg every 24 h for 2 doses) (ref: Micromedex) were designated the experimental group, while those who were not were included in the control group. This study was approved by the Institutional Review Board of Chang Gung Memorial Hospital (IRB No. 202001147B0C503), and the need for consent was waived by said IRB’s Ethics Committee.

### 2.2. Patients Studied

GA, BW, gender, number of days post-birth on initiation of ibuprofen use, hypoalbuminemia (<2.5 g/dL), G6PD deficiency incidence [13,15], receiving blood transfusion or not, phenobarbital use or not, presence of cholestasis or not, and TSB level monitored within 30 days after birth were recorded for each neonate. 

Phototherapy is the most common and effective way to treat neonates with hyperbilirubinemia, and our phototherapy devices (atom phototherapy 106 stand type, Tokyo, Japan) delivering wavelengths of 430 to 490 nm is most frequently used to convert unconjugated bilirubin [16]. The potency of phototherapy in reducing TSB is determined by TSB level at the beginning of management, the spectrum of light emitted, origin of jaundice, and uncovered body surface area [17]. Infants delivered at a younger gestational age have lower thresholds for the initiation of phototherapy [18], and initiation or suspension of phototherapy for the treatment of neonatal hyperbilirubinemia complies with “Treatment protocols using the new treatment criteria” [17]. We recorded and adjusted the influence of phototherapy on the TSB level. The phototherapy duration was documented, including the initial day and end day. Two monitored TSB levels were recorded, from which the average influence of one-day phototherapy on TSB could be determined: (TSB-TSB)/days of phototherapy. We categorized neonates based on their birth weight and gestational age, into BW < 1 kg, 1–1.49 kg, 1.50–2.49 kg, and ≥2.5 kg, and GA 21–27 weeks, 28–31 weeks, 32–33 weeks, 34–36 weeks, and ≥37 weeks, respectively.

The TSB level within 48 h prior to ibuprofen use and within seven days following ibuprofen use were recorded and compared. The highest monitored TSB level post-ibuprofen use minus the TSB level recorded prior to ibuprofen use was considered the TSB change in this study. The above-mentioned influence of daily phototherapy on the TSB level was used to correlate the results.

Neonates with any of the following conditions were excluded: those who received ceftriaxone; those with intraventricular hemorrhage (IVH) that had a significant influence in terms of increasing the BBI (bilirubin divided by birth weight index); those with TORCH (toxoplasmosis, rubella, cytomegalovirus, herpes simplex, and other organisms including syphilis, parvovirus, and varicella zoster) infections; those who died within 30 days after birth, and those with fewer than two recorded TSB levels. 

### 2.3. Statistics

All statistical analyses were performed using commercial software (SAS 9.4, SAS Institute, Cary, NC, USA). We performed a normality test using the Kolmogorov–Smirnov test. We adopted the Chi-square test or Fisher’s exact test to compare categorical variables, and the Mann–Whitney U test for comparison of continuous variables between groups. The Kruskal–Wallis test was applied to compare the reduction in TSB post-phototherapy among groups of neonates with different BWs and different GAs. One-way ANOVA was used to compare the average TSB change before and after ibuprofen use in neonates within the same BW group, and one-way ANOVA by linear contrast was used to examine trends in the TSB change among neonates within different BW groups. We also applied one-way ANOVA to compare the average TSB change before and after ibuprofen use in neonates of differing GA.

## 3. Results

In total, 151,015 neonates were born at Chang Gung Memorial Hospital (based on birth record files) during the period from January 2004 to July 2020. Of those, 382 had missing data or a BW < 400 g, 1784 were duplicates (mainly caused by having no ID at birth), five were of unknown gender, 102,473 only had their TSB monitored once, 444 died within 30 days after birth, 31,464 did not undergo phototherapy (no TSB monitoring), and 147 were infected with cytomegalovirus, herpes simplex, syphilis, or varicella zoster, were administered ceftriaxone, or experienced intraventricular hemorrhage. The aforementioned neonates were excluded, and hence a total of 14,316 neonates were identified. Among these 14,316 neonates, 97 were administered ibuprofen, while 14,219 were not. Furthermore, 33 neonates given ibuprofen were excluded owing to not having had their TSB level monitored within 48 h before or 7 days after initiation of ibuprofen use, and 792 not given ibuprofen were excluded due to not having phototherapy between two monitored TSB levels. Therefore, in total, 13,491 neonates were entered into this study, 64 who were administered ibuprofen and 13,427 who were not (Figure 1).

This study included 33 male neonates (51.6%) in the group of ibuprofen users and 7299 male neonates (54.4%) in the non-user group, and ibuprofen use was inversely correlated with BW and GA (*p* < 0.001). More ibuprofen users than non-users experienced hypoalbuminemia. (15.6% vs. 2.3%, *p* < 0.001), and the days of phototherapy (5.3–14.8 vs. 1–3, *p* < 0.001) and ratio of phenobarbital use (3.1% vs. 0.3%, *p* = 0.015) were higher in the ibuprofen users than in the non-users. We observed no significant differences between these two groups in terms of G6PD deficiency (0.0% vs. 0.7%, *p* = 1.00), blood transfusion (1.6% vs. 0.3%, *p* = 0.177), or cholestasis (0.0% vs. 0.2%, *p* = 1.00) (Table 1). We adopted the Chi-Square test or Fisher’s exact test for categorical variables and the Mann–Whitney U test for continuous variables. The average daily influence of phototherapy on the TSB level was −0.20 (−0.57~0.05) mg/dL, −0.28 (−0.84~0.13) mg/dL, −0.75 (−1.77~0.10) mg/dL, and −1.60 (−2.70~−0.50) mg/dL in neonates with a BW < 1 kg, 1–1.49 kg, 1.5–2.49 kg, and ≥2.5 kg, respectively (Table 2), indicating that neonates with a BW ≥ 1.5 kg experienced a greater reduction in TSB following phototherapy as compared with those with a BW < 1.5 kg. The average daily influence of phototherapy on the TSB level was −0.17 (−0.54~0.13) mg/dL, −0.33 (−0.96~0.18) mg/dL, −0.48 (−1.00~0.23) mg/dL, −0.95 (−2.00~0.00) mg/dL, and −1.63 (−2.70~−0.50) mg/dL in neonates with a GA of 21−27 weeks, 28−31 weeks, 32−33 weeks, 34−36 weeks, and ≥37 weeks, respectively (Table 3), suggesting that more mature neonates experienced a greater reduction in the TSB level post-phototherapy. 

The average TSB value increase following ibuprofen use was 3.38 ± 2.77 mg/dL, 2.04 ± 2.53 mg/dL, and 1.34 ± 2.24 mg/dL in neonates with a BW < 1 kg, 1–1.49 kg, and ≥1.5 kg, respectively (Table 4; owing to only three neonates having been administered ibuprofen in the group with a BW ≥ 2.5 kg, we combined this group with neonates of a BW 1.5–2.49 kg, forming a new group of neonates with a BW ≥ 1.5 kg, i.e., this table only contains three groups). Non-significant differences were found in the change of TSB level following ibuprofen use between these three groups (*p* = 0.073), but increased TSB changes with decreased neonate BW were noted post-ibuprofen use (*p* = 0.026, one-way ANOVA). We categorized GA into only three groups in Table 5: GA 21–27 weeks, GA 28–31 weeks, and GA ≥ 32 weeks, as only one neonate was in the GA 34–36 weeks group and one in the GA ≥ 37 weeks group. Table 5 shows non-significant differences in the change in TSB level following ibuprofen use between these three groups (*p* = 0.45), and a decreased neonatal GA resulted in an increased TSB change (*p* = 0.217).

## 4. Discussion

The lower the birth weight, the more significant impact ibuprofen had on the TSB level (Table 4), and very low birth weight (VLBW) (BW < 1.5 kg) neonates are at greater risk of kernicterus and brain injury due to hyperbilirubinemia [19]; as such, neonates with severe jaundice should be carefully followed while receiving ibuprofen. Extremely low birth weight (ELBW) (BW < 1 kg) neonates presented the highest TSB increase following ibuprofen use in this study at 3.38 ± 2.77 mg/dL (Table 4), and a lower TSB reduction by phototherapy was observed, at −0.20 (−0.57~0.05) mg/dL/one-day phototherapy (Table 2). In addition, aggressive phototherapy may reduce the TSB level in this group of neonates, but it increases deaths as well [20]. For preventing potential brain injury, iv acetaminophen, which has proved safe and effective for the treatment of PDA and no apparent increase of TSB, can be an appropriate alternative to ibuprofen for ELBW neonates for the treatment of PDA if they are experiencing severe unconjugated hyperbilirubinemia [21,22,23,24]. 

An increased TSB change was not correlated with a decreased neonatal GA (Table 5) as expected, indicating that GA, a term commonly used in neonates, is not consistent with the trend of BW post-ibuprofen use. The reasons commonly involved in inconsistency of GA with BW of neonates include being small for the gestational age, such as the unborn fetus did not get enough nutrients, the presence of chromosome problems, maternal multiple pregnancy, or infections, or being large for the gestational age, reasons for which include the parents being large, a diabetic mother, the mother gaining too much weight during pregnancy, or simply due to the small case number of 64 neonates (with only 61 neonates having GA).

In total, 133,937 neonates who either had their TSB level monitored once or never were excluded from this study, as we aimed to investigate the influence of ibuprofen on TSB change, and therefore required TSB monitoring at least twice in the same neonate. Those infected with any one of TORCH were excluded, as explained in previous studies of infants [25,26,27]. As most significant TSB changes occur within 30 days after birth, the TSB level was followed and recorded during this time period. Those who died within 30 days after birth were also excluded, as our objective was to explore the influence of phototherapy on TSB level and its correlation with the influence of ibuprofen on TSB level up to 30 days after birth.

Phototherapy, the use of visible light, lowers the level of TSB by converting bilirubin into water-soluble isomers that can be easily eliminated without conjugation in the liver. This method is the most common and effective way to treat neonatal hyperbilirubinemia and causes the most significant reduction in TSB level. Furthermore, phototherapy use is commonly administered to a neonate over 20 h a day, which made a comparatively equal basis for comparison in this study. Therefore, we recorded and correlated daily phototherapy use, if any, for neonates in the calculation of the influence of ibuprofen on TSB level in neonates.

Breastfeeding may slightly increase the level of TSB [28], yet we designed a study with big data involving 13,427 neonates as a comparison group (non-users) and calculated the daily influence of phototherapy on TSB, which we then used to correlate the TSB change before and after ibuprofen use in the same neonate who used ibuprofen. Hence, we assumed that breastfeeding did not have a significant impact on the results of this study. Maternal diabetes mellitus [29], maternal hypertension [29], genetic risk factors [30] and congenital hypothyroidism [31] have all been inferred to affect the development of hyperbilirubinemia in neonates. No neonate was excluded from this study due to any of these situations, as the difference in the level change of TSB was monitored and calculated in the same neonate. Therefore, we do not consider that any of the abovementioned situations played a significant role in the study results.

While protein binding is essential, this may not be the only factor related to a drug’s influence on the elevation of TSB in neonates. With a high protein binding of 99% (ref: Micromedex), ibuprofen doses used in neonates for the treatment of PDA are only 5 to 10 mg/kg/day, which are not as high a dose as for drugs that profoundly interfere with the binding of bilirubin in jaundiced neonates, such as ceftriaxone [32] or sulfisoxazole [11], with a daily dosage ranging from 50~100 mg/kg/day and 75–150 mg/kg/day, respectively (ref: Micromedex). Moles or daily dosage of a drug may also play a pivotal role in the net effect of displacing TSB, ultimately leading to significant hyperbilirubinemia or even kernicterus in susceptible neonates. Prior to measurement of free bilirubin being broadly available, TSB should be evaluated to screen for potential hyperbilirubinemia in susceptible neonates whenever a new drug is approved for use.

This study may have some biases regarding TSB change before and after ibuprofen use. We used the highest monitored TSB level within 7 days following initiation of ibuprofen use from which to deduct the one monitored within 48 h before ibuprofen in order to standardize the process and compare the TSB level increase among different groups of neonates. The TSB increases shown in Table 4 provide a clue regarding the impact of ibuprofen on neonates with different BWs, yet this result may not perfectly reflect the real influence of ibuprofen despite a regular ibuprofen dosage and a logical and fair study design, and hence we list this as a study limitation.

## 5. Conclusions

As ibuprofen use is correlated with apparent increases in TSB in neonates of a lower BW, especially in those with a BW < 1 kg, iv acetaminophen can be an appropriate alternative to ibuprofen for ELBW neonates for the treatment of PDA if they are experiencing severe unconjugated hyperbilirubinemia.

## Figures and Tables

**Figure 1 children-09-01184-f001:**
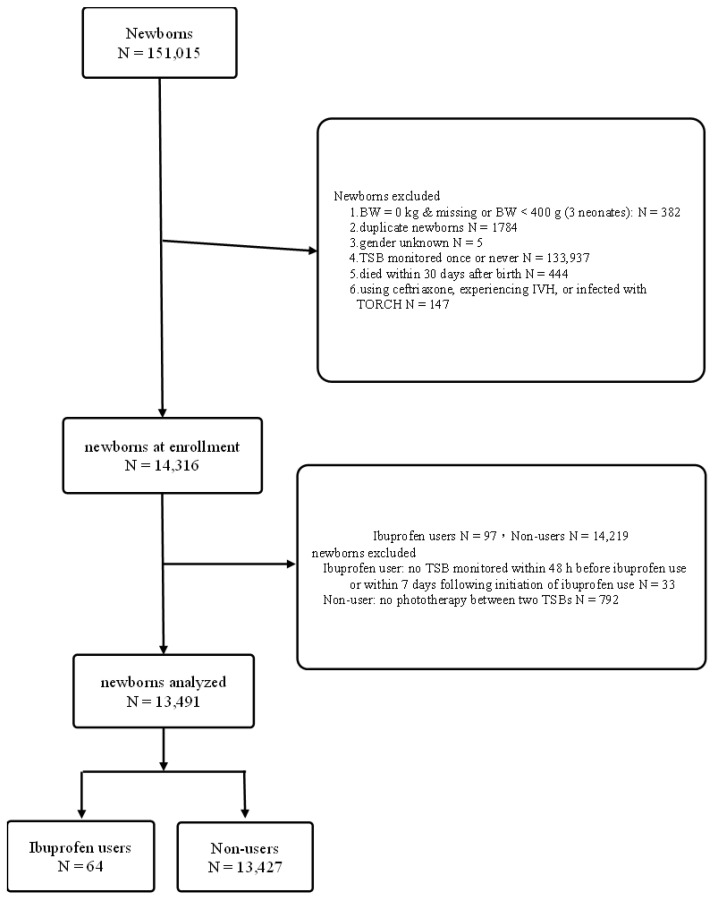
Flow chart of newborns born at Chang Gung Hospital, Taiwan, between January 2004 and July 2020, exclusions and final grouping.

**Table 1 children-09-01184-t001:** Characteristics of neonates receiving ibuprofen and none-users.

	Ibuprofen Users (*n* = 64)	Non-Users (*n* = 13,427)	*p*-Value
GA			<0.001
GA 21–27	28 (45.9 %)	414 (3.2 %)	
GA 28–31	23 (37.7%)	819 (6.3 %)	
GA 32–33	8 (13.1 %)	827 (6.3 %)	
GA 34–36	1 (1.6 %)	2055 (15.7 %)	
GA ≥ 37	1 (1.6 %)	8934 (68.5 %)	
BW			<0.001
<1 kg	16 (25.0 %)	329 (2.5 %)	
1–1.49 kg	32 (50.0 %)	668 (5.0 %)	
1.5–2.49 kg	13 (20.3 %)	2882 (21.5 %)	
≥2.5 kg	3 (4.7 %)	9548 (71.1 %)	
Gender (male)	33 (51.6%)	7299 (54.4%)	0.637
Number of days post birth for ibuprofen use	13.0 (10.0–18.8)		
Hypoalbuminemia	10 (15.6%)	303 (2.3%)	<0.001
G6PD deficiency	0 (0.0%)	96 (0.7%)	1.00
Days of phototherapy	9.5 (5.3–14.8)	1 (1–3)	<0.001
Blood transfusion	1 (1.6%)	40 (0.3%)	0.177
Phenobarbital use	2 (3.1%)	38 (0.3%)	0.015
Cholestasis	0 (0.0%)	21 (0.2%)	1.00

Chi-square test or Fisher’s exact test for categorical variables; Mann–Whitney U test for continuous variables; GA: gestational age; BW: birth weight; hypoalbuminemia: <2.5 g/dL; G6PD deficiency: glucose-6-phosphate dehydrogenase deficiency incidence; TSB: total serum bilirubin; ibuprofen users: 64 neonates with BW; 61 neonate with GA; non-users: 13,427 neonates with BW; 13,049 neonates with GA.

**Table 2 children-09-01184-t002:** Average daily influence of phototherapy on TSB levels of neonates with different BWs (Kruskal–Wallis Test).

Neonate BW	No. of Neonates	(TSB^2^-TSB^1^)/Days of Phototherapy (mg/dL)
<1 kg	329	−0.20(−0.57~0.05)
1–1.49 kg	668	−0.28(−0.84~0.13)
1.5–2.49 kg	2882	−0.75(−1.77~0.10)
≥2.5 kg	9548	−1.60(−2.70~−0.50)
Total	13,427	−1.28(−2.40~−0.20)

TSB^1^ prior to phototherapy: TSB monitored within 48 h prior to phototherapy; TSB^2^ post phototherapy: TSB monitored within 48 h after phototherapy; TSB^2^-TSB^1^: TSB monitored after phototherapy-TSB monitored prior to phototherapy.

**Table 3 children-09-01184-t003:** Average daily influence of phototherapy on TSB levels of neonates with different GAs (Kruskal–Wallis Test).

Neonates	No. of Neonates	(TSB^2^-TSB^1^)/Days of Phototherapy (mg/dL)
GA 21–27	414	−0.17(−0.54~0.13)
GA 28–31	819	−0.33(−0.96~0.18)
GA 32–33	827	−0.48(−1.00~0.23)
GA 34–36	2055	−0.95(−2.00~0.00)
GA ≥ 37	8934	−1.63(−2.70~−0.50)
Total	13,049	−1.30(−2.40~−0.20)

TSB^1^ prior to phototherapy: TSB monitored within 48 h prior to phototherapy; TSB^2^ post phototherapy: TSB monitored within 48 h after phototherapy; TSB^2^-TSB^1^: TSB monitored after phototherapy-TSB monitored prior to phototherapy. GA: gestational age; extremely premature—21–27 weeks; very premature—28–31 weeks; moderately premature—32–33 weeks; late preterm—34–37 weeks; term ≥ 37 weeks.

**Table 4 children-09-01184-t004:** Average changes of TSB levels before and after ibuprofen use in neonates with different BWs (One-way ANOVA).

Neonate BW	No. of Neonates	TSB Prior to Ibuprofen Use	Highest TSB Post Ibuprofen Use	TSB-TSB (mg/dL)
<1 kg	16	3.93 ± 1.84	5.54 ± 2.35	3.38 ± 2.77
1–1.49 kg	32	6.48 ± 2.49	6.61 ± 2.59	2.04 ± 2.53
≥1.5 kg	16	8.08 ± 2.53	8.14 ± 2.66	1.34 ± 2.24
Total	64	6.24 ± 2.76	6.72 ± 2.68	2.20 ± 2.60

TSB before ibuprofen use: TSB monitored within 48 h before ibuprofen use; TSB post ibuprofen use: highest monitored TSB within 7 days after ibuprofen use; TSB-TSB: highest monitored TSB after ibuprofen use—TSB monitored prior to ibuprofen use.

**Table 5 children-09-01184-t005:** Average changes of TSB levels before and after ibuprofen use in neonates with different GAs (One-way ANOVA).

Neonate GA	No. of Neonates	TSB Prior to Ibuprofen Use	Highest TSB Post Ibuprofen Use	TSB-TSB (mg/dL)
GA 21–27	28	5.06 ± 2.24	6.01 ± 2.29	2.59 ± 3.03
GA 28–31	23	6.68 ± 2.11	6.80 ± 2.45	2.15 ± 2.29
GA ≥ 32	10	6.87 ± 3.40	7.15 ± 3.15	1.35 ± 2.24
Total	61	5.97 ± 2.52	6.50 ± 2.50	2.22 ± 2.65

This table contains only 61 neonates due to three neonates without a GA record, different from the 64 neonates in Table 4.

## Data Availability

Not applicable.

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
