# Peer review of "Increased Total Serum Bilirubin Level Post-Ibuprofen Use Is Inversely Correlated with Neonates’ Body Weight"

_children, 2022, doi:10.3390/children9081184_

Round 1

Reviewer 1 Report

Despite the revision of the manuscript, to my opinion it has important critical limitation in form and study methods.

The period of the study is to be considered long. There could have been, among two decades, many changes in clinical practices, despite the same dosage of ibuprofen used and type of phototherapy used, that could influences the outcomes. 

It would have been more appropriated to consider only VLBW (or LBW and VLBW) or preterm newborns for this typology of study, considering that that use of Ibuprofen for PDA is rare in larger patients.

In addition, the manuscript appears very confused and difficult to read. The introduction is longer than discussion, with a lot of not usefully topic. 

From journal author guidelines “The introduction should briefly place the study in a broad context and highlight why it is important. It should define the purpose of the work and its significance, including specific hypotheses being tested. The current state of the research field should be reviewed carefully and key publications cited. Please highlight controversial and diverging hypotheses when necessary. Finally, briefly mention the main aim of the work and highlight the main conclusions. Keep the introduction comprehensible to scientists working outside the topic of the paper “

Also, the statistical methods and results presentation appears confused, with difficulty in the reading 

Line 132, maybe you mean TBS after phototherapy- TBS before phototherapy 

In an article the mention of the tables is in the results, not in the statistical section

Table 1 there are not the percentage in all variables

Sorry but to my opinion the article should be improved.

Author Response

Reviewer 1

Despite the revision of the manuscript, to my opinion it has important critical limitation in form and study methods. The period of the study is to be considered long. There could have been, among two decades, many changes in clinical practices, despite the same dosage of ibuprofen used and type of phototherapy used, that could influences the outcomes. 

A: Yes, as we mentioned last week, ibuprofen dosage is the same for all neonates until today and type of phototherapy should be the same for any given neonate.

  1. Ibuprofen has been an old drug for many decades (for neonate PDA about 20 years), and the uncertain factor now for this drug is the influence on total serum bilirubin (TSB) level in neonates, that’s what we are researching on.
  2. Phototherapy has been considered safe and effective for the treatment of hyperbilirubinemia (high TSB) also for many decades, no significant change of theory or usage. (Over 99% neonates with hyperbilirubinemia receive phototherapy)

In addition, the study period was 16.5 years via multiple centers of Chang Gung hospital (Taiwan), in total 151015 neonates, and we finally collected 97 ibuprofen users (only 64 ibuprofen users useful). If we shorten the study period then we might not be able to have enough cases for this research.

It would have been more appropriated to consider only VLBW (or LBW and VLBW) or preterm newborns for this typology of study, considering that that use of Ibuprofen for PDA is rare in larger patients.

A: The reason why we included neonates with BW <1kg, 1-1.5kg, and > 1.5kg and neonates with different GAs, because we intended to see which group of neonates (by BW or GA) are more vulnerable and easily affected by ibuprofen.

The results are 16 neonates BW< 1kg, 32 neonates BW 1-1.49, 16 neonates BW >1.5kg (Please see table 4) “Increased total serum bilirubin level post-ibuprofen use is inversely correlated with neonates’ body weight” That’s the title of this manuscript.

If we limit subjects to neonates with smaller BW, then it’s hard to make any conclusion. sorry

In addition, the manuscript appears very confused and difficult to read. The introduction is longer than discussion, with a lot of not usefully topic. 

A:OK, we trimmed unnecessary phrases in the 1,2,3, and 4 paragraphs of Introduction to make it readable and concise.

From journal author guidelines “The introduction should briefly place the study in a broad context and highlight why it is important. It should define the purpose of the work and its significance, including specific hypotheses being tested. The current state of the research field should be reviewed carefully and key publications cited. Please highlight controversial and diverging hypotheses when necessary.

A:Yes, we already have highlighted the reasons and references for this study, please see the final two paragraphs of Introduction

 Finally, briefly mention the main aim of the work and highlight the main conclusions. Keep the introduction comprehensible to scientists working outside the topic of the paper “

A1:Yes, our main conclusion now is clear  “iv acetaminophen can be an appropriate alternative to ibuprofen for ELBW neonates for the treatment of PDA if they are experiencing severe unconjugated hyperbilirubinemia” in response to your suggestion

 A2: We shorted the Introduction accordingly. (Please compare this with last version)

Also, the statistical methods and results presentation appears confused, with difficulty in the reading 

 A:OK, we took away “table” from statistics as expected

Line 132, maybe you mean TBS after phototherapy- TBS before phototherapy 

 In an article the mention of the tables is in the results, not in the statistical section

 A1: line 132 where is this sentence?

A2: OK, we corrected this mistake accordingly, all tables now only mentioned in the results

Table 1 there are not the percentage in all variables

 A: OK, we filled in appropriate percentages as expected.

Reviewer 2 Report

The manuscript "Increased total serum bilirubin level post-ibuprofen use is inversely correlated with neonates' body weight" by Lee et al. aimed to investigate the influence of ibuprofen on the increase of the total serum bilirubin level in neonates. The authors included a large number of neonates, for 16 years, based on data from the birth record files. Even though this study brings interesting data, many improvements must be made. The authors must explain clearly their aim in the abstract. Why did they analyze the efficacy of phototherapy as there was no other link with their study's aim (role of ibuprofen in increasing bilirubin level)? The tables must be included in the Results section and not in the Methods. I would move Figure 1 to the methods as it consists of the selection o the newborns. Also, the Figure must be redesigned to be smaller. All the ibuprofen users were also with phototherapy? as the non-users without therapy were excluded from the study. 

How many newborns were at risk of kernicterus with or without ibuprofen? The mean value of TSB is far from the value at risk both before and after ibuprofen administration.

In Tables 2 and 3, the authors must specify what the values from the third column represent. 

The Discussions must be improved as there is no comment about other similar studies regarding ibuprofen or acetaminophen's influence on TSB. The authors concluded that acetaminophen would be better, but there is no discussion on the possible cited proof. 

Why are there again the exclusion criteria presented here without comments?

The English language needs to be verified and corrected. Also, some errors in editing and the use of punctuation must be checked. 

The style of the references must be corrected to respect the style requested by the journal. For example, some are only with one author and "et al."; others are with "and"; others are with initials before the authors' family name; in some, the year is repeated.

Author Response

Reviewer 2

The manuscript "Increased total serum bilirubin level post-ibuprofen use is inversely correlated with neonates' body weight" by Lee et al. aimed to investigate the influence of ibuprofen on the increase of the total serum bilirubin level in neonates. The authors included a large number of neonates, for 16 years, based on data from the birth record files. Even though this study brings interesting data, many improvements must be made.

The authors must explain clearly their aim in the abstract.

A: OK. We add one sentence to make our aim clear accordingly.

Please see the abstract (in blue)

Why did they analyze the efficacy of phototherapy as there was no other link with their study's aim (role of ibuprofen in increasing bilirubin level)?

A: Phototherapy is the most common and effective way to treat neonates with

hyperbilirubinemia.(high TSB)  (ref:Burgos, A.E. Screening and follow-up for neonatal hyperbilirubinemia: a review. Clin Pediatr (Phila), 2012. 51(1): p. 7-16.)

so we had to figure out how many days of phototherapy and correlate the influence

of phototherapy with ibuprofen. (Phototherapy significantly decreases bilirubin

level)

The tables must be included in the Results section and not in the Methods.

A:Yes, all results (data) of our studies have been distributed in the Results, in total five tables. (please see that, in blue)

 And we corrected this mistake accordingly.

I would move Figure 1 to the methods as it consists of the selection o the newborns. Also, the Figure must be redesigned to be smaller.

A: OK, we have made this flow chart more concise.

All the ibuprofen users were also with phototherapy? as the non-users without therapy were excluded from the study.

A: no, they are for different diseases. Ibuprofen for treatment of PDA, phototherapy for hyperbilirubinemia. But they are related with changes of bilirubin level, with ibuprofen increase and phototherapy decrease bilirubin levels.

No-users meaning neonates did not use ibuprofen. These neonates were used for calculating average daily influence of phototherapy on the TSB level of neonates.

How many newborns were at risk of kernicterus with or without ibuprofen? The mean value of TSB is far from the value at risk both before and after ibuprofen administration.

A1: four cases out of 178 VLBW infants receiving ibuprofen developed adverse neurodevelopmental outcome. (ref 1) and kernicterus is a more serious situation that may be induced by ibuprofen use.(ref 2) Kernicterus may be associated with causes other than ibuprofen as well.

ref 1: Total serum bilirubin levels during cyclooxygenase inhibitor treatment for patent ductus arteriosus in preterm infants. Acta Paediatr. 2009 Jan;98(1):36-42.)

ref 2: Possible Ibuprofen-induced kernicterus in a near-term infant with moderate hyperbilirubinemia J Pediatr Pharmacol Ther. 2006 Oct;11(4):245-50.

A2: TSB levels in low BW infants (ref 1) or at a younger gestational age (ref 2) have lower thresholds for the initiation of phototherapy, so the commonly accepted TSB level of 5mg/dL is safe for term neonates but not safe for those BW<1kg. ( For infant with BW<1kg, TSB level<4 or even <3 mg/dL is considered safe.)

Ref 1: Muchowski, K.E., Evaluation and treatment of neonatal hyperbilirubinemia. Am Fam Physician, 2014. 89(11): p. 873-8.)

Ref 2: Romagnoli, C., et al., [Physiologic hyperbilirubinemia in low birth weight newborn infants: relation to gestational age, neonatal weight and intra-uterine growth]. Pediatr Med Chir, 1983. 5(5): p. 299-303.

 In Tables 2 and 3, the authors must specify what the values from the third column represent.

A: Thanks for your advice. We add numbers 1 and 2 to make table 2 and 3 clearer. (in blue)

The Discussions must be improved as there is no comment about other similar studies regarding ibuprofen or acetaminophen's influence on TSB. The authors concluded that acetaminophen would be better, but there is no discussion on the possible cited proof.

A1: iv acetaminophen has proved safe and effective for the treatment of PDA, and no apparent increase TSB level (ref 1,2). Please see 1st paragraph of Discussion. (in red)    

Ref 1 Marconi, E., et al., Efficacy and safety of pharmacological treatments for patent ductus arteriosus closure: A systematic review and network meta-analysis of clinical trials and observational studies. Pharmacol Res, 2019. 148: p. 104418.

Ref 2 Dani, C., et al., Intravenous paracetamol in comparison with ibuprofen for the treatment of patent ductus arteriosus in preterm infants: a randomized controlled trial. Eur J Pediatr, 2021. 180(3): p. 807-816.

A2: OK, we rephrase the sentence as “iv acetaminophen can be an appropriate alternative to ibuprofen for ELBW neonates for the treatment of PDA if they are experiencing severe unconjugated hyperbilirubinemia” at the abstract, first paragraph of Discussion, and conclusion. (in red)

Why are there again the exclusion criteria presented here without comments?

A: yes, we already had description. Please see the 3rd paragraph of Discussion, and the 4th paragraph of patient studied of Materials and methods. (in blue)

The English language needs to be verified and corrected. Also, some errors in editing and the use of punctuation must be checked.

A:OK

The style of the references must be corrected to respect the style requested by the journal. For example, some are only with one author and "et al."; others are with "and"; others are with initials before the authors' family name; in some, the year is repeated.

A: OK, we have corrected.

Round 2

Reviewer 1 Report

.

Author Response

As the study include the role of phototherapy in decreasing TSB, the authors need to present in the Methods or Results section how they use these data in analyzing the influence of ibuprofen. 

Still, changes must be made regarding the editing of the References list. 

A1: Yes, we already have this description. Please see the 2 and 3rd paragraph of Patients studied of Materials and methods. (in blue)

A2: OK, no additional reference involved

Reviewer 2 Report

The authors improved the manuscript based on some of the comments.

As the study include the role of phototherapy in decreasing TSB, the authors need to present in the Methods or Results section how they use these data in analyzing the influence of ibuprofen. 

Still, changes must be made regarding the editing of the References list. 

Author Response

As the study include the role of phototherapy in decreasing TSB, the authors need to present in the Methods or Results section how they use these data in analyzing the influence of ibuprofen. 

Still, changes must be made regarding the editing of the References list. 

A1: Yes, we already have this description. Please see the 2 and 3rd paragraph of Patients studied of Materials and methods. (in blue)

A2: OK, no additional reference involved

This manuscript is a resubmission of an earlier submission. The following is a list of the peer review reports and author responses from that submission.

Round 1

Reviewer 1 Report

Major comments

  1. First of all congratulations for your work and your efforts to investigate an argument still controversial, and of interest in neonatology. However, my opinion is that your paper cannot be accepted for pubblication in its current form because, to me, the methodology of the study is not fully adequate, and the conclusions provided are not based on enough evidences. I will try to address my arguments below in the following points.

  2. In the conclusions, you state that “Since ibuprofen use is correlated with apparent increases of TSB in neonates with lower BW, especially in those with BW<1 kg, we recommend ELBW neonates be prescribed acetaminophen instead of ibuprofen for the treatment of PDA if they are experiencing severe unconjugated hyperbilirubinemia.”.
    To me this recommendation is not supported by enough evidences (the results of a single centre retrospective study) and could even represent a danger if adopted in patients with hemodinamically significant patent ductus arteriosus (PDA).
    It is my opinion that, when evaluating the risk/benefit balance of PDA treatmet in very preterm infants, along with the echographic and clinical characteristic of the neonate, should be also considered:

    • That current recent RCTs have suggested that rate of PDA closure with IV acetaminophen is significantly lower than with either indomethacin or ibuprofen [Davidson JM, et a.l A J Perinatol 2021;41(1):93-99. Dani C, et al. Eur J Pediatr 2021;180(3):807-16. Liebowitz M, et al. J Perinatol 2019;39(5):599–607.] and that the most recent guidelines recommend that Ibuprofen should be considered the pharmacotherapy of choice for a symptomatic PDA (strong recommendation).[Mitra S, et al. Management of the patent ductus arteriosus in preterm infants. Paediatr Child Health. 2022 Mar 7;27(1):63-64.].

    • That PDAs have been associated with numerous adverse outcomes, including prolongation of assisted ventilation, pulmonary hemorrhage, chronic lung disease (CLD), necrotizing enterocolitis (NEC), intraventricular hemorrhage (IVH), and death [Benitz WE. Pediatrics 2016;137(1); Mitra S. Clin Perinatol 2020;47(3):617-39.]

Moreover, from the manuscript it is not possible to understand if those higher TSB values found in patients after Ibuprofen treatment, led to a more complicated clinical course (e.g. higher number of days of phototherapy, higher number of exchange-transfusions performed, among the others) or to worst short-term outcomes.

For these reasons, to me, in the conclusions, it could have been more appropriated to suggest to the readers to consider also the possible risk of increased TSB values when evaluating the risk/benefit balance of Ibuprofen treatement of PDA in smaller patients, until stronger evidences would be available.

  1. Even if it seems very clear from the title and the introduction that the main aim of the study was to “determine how ibuprofen use influences TSB levels of neonates with different birth weights (BW) and gestational ages (GA)” (and this refer mainly to preterm infants where this drug is used), a substantial part of the manuscript describes the effect of phototerapy on daily TSB reduction (in term infants). To me, the second one is a completely different research topic, regarding a different population and deserve to be treated in a separate report. Moreover, If I have fully understood the meaning of “average daily influence of phototherapy on TSB levels of neonates “, results of your study seems to show that phototerapy is almost not effective, being able on average to reduce TSB values less than 1 mg/dL per day in preterm infants.

  2. To me, it would have been more appropriated to consider only VLBW (or LBW and VLBW) for this study, considering that that use of Ibuprofen for PDA is rare in larger patients.

  3. The period of the study (2004-2020) is for me too be considered long. There could have been, among nerly two decades, many changes in clinical practices, dosage of ibuprofen used and type of phototherapy used.

  4. In the methods are missing references to type of lamps used for phototerapy and if were used single, double or triple phototerapy. Also references to criteria for treatment of hyperbilirubinemia are missing.

  5. In the methods are not reported (meanwhile present in the conclusions) the dosages used for treatment with Ibuprofen.

  6. To me, it would have been more appropriated to use a control group of preterm who did not received Ibuprofen, matched for BW; GA and clincal charachteristics, to compare the TSB values after 7-10 days with those of patients who received Ibuprofen.

  7. Among the characteristics of patients, when investigating hyperbilirubinemia in neonates, should have been reported the number of infants with a positive direct antiglobulin (DAT) test, also known as Coombs test.

Minor comments

  1. The phrase "Hyperbilirubinemia is caused by bilirubin deposition in the skin" (in: 1.Introdcution , line 47) it is not correct and confounding. Even the following part of the phrase "and most jaundice in newborns is a result of increased red blood cell breakdown and decreased bilirubin excretion" it seems inadequate to me, seeming to refer more to hemolitic diseases and not being able to explain why preterm neonates are more at risk of hyperbilirubinemia.

  2. In the introduction, when explaining hyperbilrubinemia in newborn, there is no mention to the distinction between conjugated and unconjugated bilirubinemia. Moreover, other common causes of hyperbilirubinemia (i.e hemolitic disease of the newborn and other causes) should also have been reported.

  3. The phrase "Bilirubin displacement ability has been considered an important factor for a drug to displace bilirubin, subsequently leading to hyperbilirubinemia in a neonate." that can be found both in the abstract and in the introduction (lines 19 and 77) it is not very clear to me.

Reviewer 2 Report

The Authors performed a retrospective study, to evaluate how ibuprofen use influences total serum bilirubin levels of neonates with different birth weights and gestational ages.

Despite the beautiful and interestly idea, and the novelties presented in this study, I read with difficulty the work. It appears not well written and confused. There is the reference of figure not reported (where is the flow chart?). In addition, conclusions appears very strong: "we recommend ELBW neonates be prescribed acetaminophen instead of ibuprofen for the treatment of PDA if they are experiencing severe unconjugated hyperbilirubinemia". You founded that ibuprofen use is correlated with apparent increases of TSB in neonates with lower BW, especially in those with BW <1 kg but not explored the side effects of other drugs, maybe more caution should be used. Also the abstract appears confused.

I'm very interested in the study results, however I suggest to revise the English of the study and the results presentation and to resend the manuscript